# Enhanced Stability of Vegetal Diamine Oxidase with Trehalose and Sucrose as Cryoprotectants: Mechanistic Insights

**DOI:** 10.3390/molecules28030992

**Published:** 2023-01-19

**Authors:** Meriem Megoura, Pompilia Ispas-Szabo, Mircea Alexandru Mateescu

**Affiliations:** Department of Chemistry, Faculty of Science, Research Chair on Enteric Dysfunctions “Allerdys” & CERMO-FC Center, Université du Québec à Montréal, C.P. 8888, Branch Centre-Ville, Montréal, QC H3C 3P8, Canada

**Keywords:** histamine, enteric dysfunction, vegetal diamine oxidase, freeze-drying, gastro-protection, stability studies, sucrose, trehalose

## Abstract

Enteric dysfunctions are common for various histamine-related intestinal disorders. Vegetal diamine oxidase (vDAO), an enzyme able to decompose histamine and thus alleviate histamine-related dysfunctions, was formulated in gastro-resistant tablet forms for oral administration as a food supplement and possible therapeutic agent. A major challenge for the use of proteins in the pharmaceutical field is their poor stability. In this study, vDAO was freeze-dried in the absence or in the presence of sucrose or trehalose as cryoprotectants and then formulated as tablets by direct compression. The stability of the obtained preparations was followed during storage at 4 °C and −20 °C for 18 months. In vitro dissolution tests with the vDAO powders formulated as tablets were performed in simulated gastric and in simulated intestinal fluids. The tablets obtained with the powder of the vDAO lyophilized with sucrose or trehalose cryoprotectants offered better protection for enzyme activity. Furthermore, the release of the vDAO lyophilized with the cryoprotectants was around 80% of the total loaded activity (enzyme units) compared to 20% for the control (vDAO powder prepared without cryoprotectants). This report revealed the potential of sucrose and trehalose as cryoprotectants to protect vDAO from freeze-drying stress and during storage, and also to markedly improve the vDAO release performance of tablets obtained with vDAO powders.

## 1. Introduction

Various pathological conditions, such as gut mastocytosis [1], severe falciparum malaria [2], and inflammatory bowel disease [3], recognized as global diseases, are associated with high levels of plasma histamine. In this context, the role of histamine at the level of the intestinal mucosa is of particular awareness. Acting as a “*friend and foe”*, histamine plays a protective role in maintaining a healthy gut [4] but also produces severe dysfunctions, such as the enhancement of intestinal permeability, allergic symptoms, altered host-microbiota immune interactions, and disruption of homeostasis, that can lead to severe bowel disorders. The intestinal microbiome is composed of up to 1000 different bacterial species [5]. A large range of bacterial strains in the human gut can produce histamine, generating an imbalance of the histamine level. Intestinal histamine-producing bacteria could enhance histamine accumulation in the colon, followed by its absorption into the plasma and the development of several related diseases [6]. Furthermore, high levels of histamine are also present in some fermented food [7], wine [8], scombroid fish, and some vegetables [9], leading to food intoxication (histaminosis) and frequently related pseudo-allergies [10].

Vegetal diamine oxidase (vDAO) was proposed as an oral enzymatic therapy to treat histamine-related intestinal dysfunctions [11,12,13]. Diamine oxidase (EC 1.4.3.22) is a copper amine oxidase [14] that catalyzes the oxidative deamination of primary amines, such as histamine, tyramine, and cadaverine [15], into corresponding aldehydes, ammonia, and hydrogen peroxide (from the concomitant reduction of molecular oxygen) [16]. The enzyme is constituted of dimers of identical subunits from 70 to 90 kDa, each containing a single copper ion and a cofactor covalently bound and formed by the post-translational modification of a tyrosine side chain to 2,4,5-trihydroxyphenylalanine quinone (Topaquinone, TPQ) [14,17,18]. The DAO was mainly produced and commercialized using pig kidneys (Porcine kidney DAO; pkDAO). More recently, vDAO from *Pisum sativum* supplementation was proposed in capsule and tablet forms. Exogenous DAO supplementation has improved histamine-related symptoms in clinical studies [19]. Compared to DAO from animal origin (pkDAO), the vDAOs were found to have higher activity in terms of enzyme units [20,21] and are safer in terms of side effects. In addition, some sociocultural requirements may limit the consumption of pig components in dietary supplements [22]. DAO from *Lathyrus sativus* was also reported to reduce histamine toxicity in vitro and was shown to alleviate symptoms in patients suffering from histamine sensitivity [23].

vDAO is currently obtained as an extract, and the enzyme should be dried to obtain oral dosage forms (tablets obtained by direct compression of powders) for oral administration. Proteins in the solution may have physical and chemical instability related to aggregation, denaturation, solubilization, oxidation, deamidation, and hydrolysis [24,25]. Protein stability is a particularly relevant aspect in the pharmaceutical field and will continue to be more significant as more therapeutic protein products are under investigation [25]. Freeze-drying, or lyophilization, is a common technique in developing solid-state therapeutic proteins [26,27,28]. The process consists of three steps: freezing the protein solution, sublimating the water in the primary drying (ice sublimation), and removing additional moisture by secondary drying [28]. During lyophilization, the proteins are subjected to various types of stress (the rate of temperature change, osmotic pressure, and pH changes) [29]. The proper choice of excipients and process controls are critical to protecting the proteins during the freeze-drying process. Various cryoprotective excipients, including disaccharides, polyols, and amino acids, can be added before freeze-drying [30,31]. Among these, trehalose (TRE) and sucrose (SUC) disaccharides are often used as stabilizers to protect the proteins against the stress of drying and during storage [32,33,34,35]. It was hypothesized that these sugars might substitute the water around the polar residues during drying, preventing the destabilization of macromolecular assemblies. Consequently, the proteins may preserve certain stability, even without certain amounts of water [36]. The prominent mechanistic hypotheses for the phenomena occurring during freeze-drying are: (i) water substitution and (ii) vitrification (glass dynamics hypothesis) [30,32,33,37]. The concept of water replacement (i) can be classified as a “specific interaction mechanism” [35], stating that stabilizers can form hydrogen bonds at specific sites on the protein surface and replace the thermodynamic stabilization function of water lost at drying [30]. The vitrification theory (ii) suggests a stabilizer matrix able to induce a kind of immobilization of the protein structures [33]. Vitrification prevents undesirable protein–protein interactions and protects the native structure of the proteins from damage during lyophilization [36]. The kinetic stability of the vitrification process is described by the characteristic glass transition temperature, T_g_, of the glassy matrix comprising the amorphous stabilizer and the protein [24,37]. The stabilization of the protein structure during lyophilization does not ensure its stability during long-term storage [38]. It is recommended that the storage temperature should be at least 20 degrees below the glass transition temperature of the solid amorphous matrix [39]. The glass transition temperature is not the only parameter affecting the effectiveness of cryoprotectants. It was suggested that residual moisture might have a decisive influence on the long-term stability of the proteins [39,40].

This report is aimed to evaluate the stability of vegetal diamine oxidase during freeze-drying, the long-term storage of powders and tablets, and the behavior of vDAO-tablets obtained with vDAO extracted from *Pisum sativum* seedlings lyophilized with two different disaccharides as cryoprotectants. The morphologic properties, water content, and T_g_ of the freeze-dried powders were assayed by scanning electron microscopy (SEM), thermogravimetric analysis (TGA), and differential scanning calorimetry (DSC), respectively. The enzyme catalytic activity during the storage period was used as the control parameter to evaluate the preparations in both powder and tablet forms.

## 2. Results and Discussion

### 2.1. vDAO Activity after Lyophilization

The vDAO activity was measured before and after lyophilization. The remaining activity of the freeze-dried powders was 75% for the control and slightly higher with the sucrose at 81% and 83% with the trehalose (Figure 1). The protection was independent of the type of disaccharide.

In this study, the activities of vDAO formulated either with sucrose (SUC) or with trehalose (TRE) were almost similar, with a slight difference of 2%. Similar results were observed in previous studies on protein lyophilization with sugars. For instance, xanthine oxidase [41] and lactate dehydrogenase [27,42] maintained good activities, and BSA (Bovine Serum Albumin) [43] preserved its structural features after freeze-drying in the presence of disaccharides as cryoprotectants. These results may be attributed to optimal lyophilization parameters and to the chemical and physical properties of the disaccharides, such as a high T_g_ and molecular mobility below its T_g_ [27] and their ability to form hydrogen bonds with the protein during the freeze-drying process.

### 2.2. Thermal Properties of the Freeze-Dried Formulations

The moisture content of the freeze-dried formulations was calculated at 105 °C from thermogravimetric analysis up to 300 °C. The DSC analysis allowed the determination of glass transition temperatures. The thermal properties of the lyophilized samples are summarized in Table 1.

The vDAO (control) showed crystallization at 152 °C (about 110 °C above the T_g_ value), whereas the formulations with the disaccharides as cryoprotectants remained amorphous (no crystallization was observed in the studied temperature range). With a similar moisture content in the presence of the cryoprotectants, the formulations containing trehalose had a higher T_g_ value than those containing sucrose. Among the disaccharides, trehalose has been reported to have a T_g_ higher than that of sucrose [44]. Some studies reported that sucrose tended to crystallize before trehalose [41]. In the case of the xanthine oxidase samples freeze-dried with sucrose, the Tc was 132 °C, within the same temperature range. Considering that vDAO is a glycoprotein [45], it is possible that disaccharide cryoprotectants could also interact with the carbohydrate moiety of the vDAO and prevent crystallization. It is also worth noting that the moisture content of the vDAO without cryoprotectants was higher compared to the samples with sucrose or trehalose. Previous studies have shown negative consequences at high moisture and temperature values on the structural integrity and stability of freeze-dried biologics [46,47]. The increase in the moisture content can reduce the T_g_ due to its plasticizing effect [48]. The presence of cryoprotectants provides a smooth appearance, with skeletal structures to the lyophilizates. It is supposed that this morphology could be the key to the low residual humidity in freeze-dried samples with sucrose or trehalose and thus protects the vDAO against the decrease of enzyme activity during storage time.

### 2.3. Characterization of vDAO Powders

The morphology of the lyophilized samples with or without cryoprotectants agents (CPAs) was characterized by SEM (Figure 2).

The vDAO lyophilizate without cryoprotectants has a very porous appearance (Figure 2A), clearly differing from the vDAO lyophilized in the presence of sucrose (Figure 2B) or of trehalose (Figure 2C), with both showing smoother surfaces. The freeze-dried disaccharides lyophilized alone under the same conditions (control) also exhibited a smooth appearance with “skeletal” structures (Figure 2D,E). The freeze-drying of vDAO in the absence of cryoprotectants significantly increases the number of pores on the surface of the dry enzyme. After the solidification of all the solutes and water, the temperature of the platform increases, providing energy for sublimation, and the ice vapor is evacuated; thus, primary drying begins. The removal of the ice crystals by sublimation would probably create an open network of pores, which are pathways for further advanced water removal from the vDAO. The freezing step can affect the morphological characteristics of the freeze-dried cake [49], such as the size of the ice crystals, the surface of the proteins, and the porosity of the final cake (Figure 2A). In the presence of disaccharide cryoprotectants, the smooth, non-porous appearance of the vDAO is probably due to the interaction of the disaccharides:disaccharides or disaccharides:protein during lyophilization.The enzyme appears to be incorporated in the predominant carbohydrate matrices. Imamura et al. [43] reported that when a protein solution is freeze-dried in the presence of sugars, amorphous matrices of the sugars are formed, which could integrate protein macromolecules [43]. The incorporation of protein molecules into amorphous sugar matrices may prevent the conformational changes in the protein during dehydration and prevent water access during storage due to the interactions between the protein and sugar molecules [43,50].

### 2.4. vDAO Activity after Long-Term Storage at Different Temperatures

A protein-to-sugar weight ratio of 1:1 is suitable for optimal stabilization [51]. In this study, the enzyme:sugar ratio (calculated as *w*/*w* mg protein:mg disaccharide) was intentionally increased to 1:10 to prevent protein collapse during lyophilization [39].

The residual humidity of the freeze-dried samples was calculated at 105 °C from the sample weight measurements taken after thermogravimetric analysis during the storage period (18 months) at 4 °C and −20 °C (Figure 3).

After 18 months of storage, the vDAO samples without disaccharides (control) showed a significant increase in moisture content, reaching 16% and 17% at 4 °C and −20 °C, respectively. The moisture content may be attributed to the eventual remaining traces of salts (i.e., phosphates) used during the extraction procedure of the vDAO, which can retain humidity. The vDAO is a glycoprotein with an N-linked carbohydrate content (3–14% in vegetal DAOs) [52,53]. The carbohydrate moiety of the enzyme can be destabilized during drying and storage, and this impacts the conformation of the vDAO and decreases its efficiency.

The presence of sucrose or trehalose as CPAs significantly reduced the moisture content, allowing the preservation of a low RH content of the vDAO formulations. This could be explained by the interaction of the disaccharides with the protein structure (possibly via the carbohydrate moiety) by hydrogen bonds, which stabilize the enzyme conformation and have a positive impact on the preservation of vDAO activity.

The values of vDAO activity after 18 months of storage (Figure 4) showed a gradual decrease at 4 °C. Practically, the enzyme without cryoprotectants retained only 43%, whereas the remaining enzyme activity for both formulations containing cryoprotectants was 60% at the end of the storage period. The remaining activities for the samples stored at −20 °C were around 40% for three formulations (the vDAO with or without CPAs). Unexpectedly, the activity of the enzyme was higher at 4 °C than at −20 °C at 18 months. This can be explained by the storage temperatures being lower than the T_g_ of the samples. Keeping the proteins stored below their T_g_ is important but not sufficient to ensure long-term enzyme stability [54].

In the absence of a cryoprotectant agent, the protein conformation can be damaged during the lyophilization process. During the secondary drying, the excess moisture is removed, including the water from the surface of the protein, and this can destabilize the protein conformation, making it more sensitive to environmental factors. In fact, at 4 °C and −20 °C, the moisture content of the vDAO powder lyophilized without CPAs reached 16%, meaning that the interaction of the enzyme with water affects its stability during long-term storage (Figure 3). Adding the CPAs reduced the moisture content in both formulations (with sucrose or trehalose) and maintained 70% of the total activity after a storage period of 18 months at 4 °C. This value was higher than the retained activities at −20 °C.

### 2.5. Behavior of the Enzyme Formulations in Tablet Forms

#### 2.5.1. Stability of vDAO in the Dry Tablets

Tablets with DAO powders formulated with sucrose or with trehalose as cryoprotectants were prepared and their physical properties are summarized in Table 2. Formulations based on the vDAO and vDAO lyophilized with sucrose generated harder tablets than those formulated with trehalose.

The stability of vDAO activity in the dry tablets was also investigated by measuring the enzymatic activity (Figure 5) after one, three, and six months of storage at 4 °C, which appeared to allow better stability of the vDAO powders (Section 3.4).

The activity of the vDAO decreased gradually, but the loss of activity was lower in the case of the tablets obtained with powders of vDAO lyophilized with disaccharides as the cryoprotectants. After 6 months of storage at 4 °C, the remaining activity was 76% for the tablets of the vDAO freeze-dried with sucrose and 71% for the tablets of the vDAO freeze-dried with trehalose, whereas the activity of the vDAO freeze-dried without cryoprotectants (control) was 64% of the initial activity. The decrease in enzyme activity may be related to moisture absorption. A decrease in the hardness of the tablets was also observed for the mentioned period (the data are not presented).

#### 2.5.2. In Vitro Dissolution Tests

The dissolution patterns of the various vDAO preparations (freeze-dried with and without cryoprotectants) formulated in tablet forms, in simulated gastric fluid (SGF), and in simulated intestinal fluids (SIFs) are presented in Figure 6.

The tablets were incubated first in the SGF for two hours (the time recommended by the US Pharmacopeia to simulate stomach transit). No release of vDAO activity was detected during the 2 h of incubation in the SGF. The tablets were then transferred into the SIF, where the tablets containing vDAO-SUC and vDAO-TRE were released in a controlled manner at 80% and 70%, respectively, of the total loaded enzyme activity (in terms of enzyme units) after 5 h of incubation (2 h in the SGF and 3 h in the SIF), showing good protection of the vDAO against the gastric acidity and the intestinal proteolytic enzymes (pancreatin), whereas from the tablets obtained with the cryoprotectants-free vDAO powder, only 20% of the loaded enzymatic activity was found in the dissolution medium. During the 6 months of storage, the dissolution profiles (Figure 6) showed the same trends of enzyme release patterns: 80% of vDAO activity was found for the tablets with vDAO:SUC and 70% for vDAO:TRE, whereas 25% of activity was found for the tablets loaded with vDAO (control).

The vDAO activity was better preserved in tablet form than in powder form. This was probably due to the excipients used for the tablet preparation, which provided extra protection to the enzyme during storage. It is known that the enzymatic activity is dependent on the pH of the surrounding media [11,22]. The fact that there was no vDAO detected in the SGF can be related to the carboxymethylcellulose (CMC) excipient. In gastric acidity, the carboxylic groups of CMC can be protonated, forming an outer compact gel preventing the liberation of the vDAO, as well as the gastric acid entering the tablet and the enzyme denaturation.

The presence of the SUC or TRE cryoprotectants not only protected the vDAO during the lyophilization and the storage, but their presence in the formulation of the vDAO tablets allowed a better release of the enzyme than the control (vDAO lyophilized without CPAs). The high amount of 70 and 80% vDAO released from the tablets formulated with the enzyme freeze-dried with the cryoprotectants is of interest, showing that the sucrose and trehalose may act as modulators of vDAO release. The slightly higher percentage of vDAO liberation from the tablets with vDAO:SUC, when compared to tablets with vDAO:TRE, can be explained by the higher solubility of sucrose (200 g/100 mL) compared to that of trehalose (68.9 g/100 mL). Consequently, sucrose may behave as a kind of “swelling” accelerator, favoring the enzyme’s liberation. In addition, the release time of 5 h is optimal, considering the 5–6 h transit time of an orally taken tablet to reach the colon.

The gastrointestinal tract is considered a difficult environment for the enzyme due to the effects of the pH and proteolytic enzymes (pepsin and pancreatin). A previous study shows that vDAO is sensitive to pH variations and could be inactivated in harsh environmental conditions [22]. The additional carbohydrate matrices offered by the CPAs may provide additional protection of the vDAO activity in simulated gastric and intestinal fluids by limiting the direct contact between the enzyme and the external medium (i.e., the exposure to the proteolytic enzymes, such as intestinal trypsin and chymotrypsin), thus allowing a higher release of the formulated enzyme in the tablets. This may be explained by the fact that the presence of sucrose and trehalose allows more flexibility for the formation of hydrogen bonds with the protein. Due to these more flexible bonds, sucrose or trehalose may interact with the surface of the enzyme and with the external media (gastric or intestinal) at the same time [55].

#### 2.5.3. Commercial products

Two commercial vegetal diamine oxidases as food supplements in tablet and capsule forms were also evaluated for enzyme release under the same conditions as previously indicated for the tablets in this study (2 h in the SGF, then transferred into the SIF until complete dissolution). After 1 h in the SGF, the commercial tablets were fragmented, and the capsules lost their integrity (Appendix A). The fact that the commercial capsule and the tablet lost their integrity in the dissolution medium suggests that the enzyme release cannot be controlled during gastric transit. The enzyme activity of both the capsule and tablet in the SGF was ≤10%, which could be related to an inactivation of the enzyme in the gastric medium. When exposed to the SIF, the maximal enzyme activity for the tablet reached 20% after 1h and then decreased to 8% at the end of the test, whereas the capsules released 50% of their activity in 3 h (Figure 7). Compared to the monolithic tablets here described with the obtained freeze-dried vDAO powders, the commercial tablets did not offer the same level of gastroprotection. The results obtained using vDAO and cryoprotectants confirmed the capacity of the disaccharides to protect the enzyme during the lyophilization process and, in association with the excipients, to better protect against the pH variations in different dissolution media.

## 3. Materials and Methods

### 3.1. Materials

Vegetal diamine oxidase from *Pisum sativum* (Diamaze 1.2 U/mg protein) was from IBEX Pharmaceuticals Inc. (Montreal, QC, Canada) and kept at −80 °C. Monobasic and dibasic phosphate, 4-Aminoantipyrine (AAP), Sodium 3,5-dichloro-2-hydroxybenzenesulfonate (DCHBS), horseradish peroxidase (HRP), Putrescin, Ammonia assay kit, Bovine Serum Albumin (BSA), sucrose, trehalose, carboxymethyl cellulose (CMC), Mg stearate, pepsin from porcine mucosa (460 units/mg solid), and pancreatin from porcine pancreas (8X USP specifications) were purchased from Sigma Aldrich (Oakville, ON, Canada). The Bradford reagent was purchased from BioShop^®^ (Burlington, ON, Canada). The Hydroxypropyl Methyl Cellulose (HPMC) K100 was gifted by Colorcon (Harleysville, PA, USA).

### 3.2. Preparation of Enzyme Formulation and Lyophilization

The vDAO was thawed overnight at 4 °C. The electrophoretic zymographic pattern of enzyme activity and the SDS electrophoretic pattern of proteins (Appendix A) were performed, as described by Chomdom Kounga et al. [56], to complete the characterization of the received product. The vDAO was then treated with cryoprotectants (sucrose or trehalose). The ratio of vDAO:cryoprotectant was 1:10 (*w*:*w*). Once mixed, the solutions were divided into 50 mL plastic tubes and maintained at −80 °C for 24 h. After freezing, the tubes were immersed in liquid nitrogen for 2 min before lyophilization. The preparations were then freeze-dried on a benchtop Virtis Freezemobile 25EL (New York, NY, USA) with a chamber pressure of 2.5 × 10^−2^ Torr, increasing the temperature from −86 °C to 25 °C 72 h. All obtained powders were stored at 4 °C and −20 °C, and their stability was tested after 1, 2, 3, 6, 12, and 18 months.

### 3.3. vDAO Activity Measuring

Specific activity: DCHBS-AAP-HRP method

The vDAO catalytic activity was assayed by measuring the rate generation of H_2_O_2_ during the oxidation of putrescine by the DCHBS-AAP-HRP method, as reported [57]. Briefly, in a spectrophotometric cuvette at 25 °C and following the absorbance variation at 515 nm, vDAO samples were diluted in 1 mL of a phosphate buffer, 0.1M pH 7.4, with AAP (0.1 mM final concentration), DCHBS (0.1 mM final concentration), horseradish peroxidase (3.6 U/mL final concentration), and putrescine (3 mM final concentration) then added. The role of AAP is to form a pink complex with the oxidized DCHBS. The latter was oxidized in the presence of horseradish peroxidase by H_2_O_2_ generated by the oxidative deamination of putrescine by vDAO. For the rate of H_2_O_2_ calculation, a molar extinction coefficient of 26,000 was used. The vDAO activity was expressed in enzymatic units (U); 1 unit corresponds to 1 μmol of H_2_O_2_ generated/min.

2.Protein quantification

Protein concentrations of vDAO were determined by a Bradford assay [58] using bovine serum albumin as the standard, and the absorbance was monitored at 595 nm.

### 3.4. Scanning Electron Microscopy (SEM)

Scanning electron microscopy images of the lyophilized powders were obtained with a scanning electron microscope, model S-3400N type II (Hitachi High Technologies America, Pleasanton, CA, USA); the secondary electron detector was used for high-resolution images. The samples were mounted on SEM sample holders with double-stick carbon tape. The mages of the samples were obtained in a high-vacuum SEM mode with a 10 keV electron beam.

### 3.5. Thermogravimetric Analysis (TGA)

The weight loss, depending on the variation of temperature, was measured by a Q500 TGA Thermogravimeter (TA Instruments, New Castle, DE, USA) between 30 and 300 °C at 10 °C/min, using a small amount (1 to 5 mg) of powders filed on a platinum crucible. To keep a controlled atmosphere, the samples were maintained under nitrogen at a flow rate of 100 mL/min. The water content was determined at about 105 °C. All the results were treated by “TRIOS” software version 4.4.0.41651.

### 3.6. DSC (Differential Scanning Calorimetry) Measurements

Differential scanning calorimetry (DSC) analyses of the freeze-dried formulations were performed using a DSC 1 STARe System (Mettler Toledo, ON, Canada). Approximately 10 mg of the lyophilized samples were weighed in an aluminum pan (40 µL) with a pierced lid and sealed hermetically. An empty pan was used as a reference. The experiments were conducted from 0 to 180 °C at a scan rate of 5 °C/min under a nitrogen atmosphere. DSC thermograms were analyzed by STARe evaluation software version 16.20 interfaced with the DSC.

### 3.7. Preparation of Monolithic Tablets

Monolithic tablets of 300 mg with 10% of vDAO powders previously lyophilized and stored at −20 °C were prepared by mixing the excipients: sodium carboxymethyl cellulose (NaCMC), magnesium stearate, and hydroxypropyl methylcellulose (HPMC). The excipients were initially homogenized manually in a mortar for 2 min, then transferred into 50 mL tubes and mixed for 10 min in a rotary device at 50 rpm. Dry powders of all excipients and dry vDAO were mixed for each formulation in quantity equivalent to 20 tablets. Tablets were obtained by direct compression of the dry powders in a 9 mm cylindrical mold with flat punches using a manual hydraulic press (Wabash, IN, USA) at 2 tons for 10 s. All the tablets were then packed in HDPE (High-Density Polyethylene) bottles without desiccants and stored at 4 °C for stability studies.

#### 3.7.1. Characterization of Dry Tablets

Tablet hardness was evaluated with a VK 200 hardness tester (Benchsaver™ Series, Vankel, NC, USA), as described in the United States Pharmacopeia (USP) <1217>. The thickness of the tablet was evaluated with a caliper from DOCAP tools (St-Laurent, QC, Canada).

The enzymatic activity of vDAO in dry tablets was measured after crushing them in a mortar and then suspending the material in 10 mL of a phosphate buffer (0.1 M; pH 7.4). The suspension was then placed in an incubator shaker for 1 h at 37 °C at 100 rpm to allow the maximum release of the enzyme. The vDAO solutions were filtered on filter paper (FisherBrand filter paper, medium porosity); the vDAO activity was determined as described in Section 3.3.

#### 3.7.2. In Vitro Dissolution Tests

The tablets’ behavior was followed in simulated gastric fluid (SGF) for 2 h and intestinal fluid (SIF) until complete dissolution of the tablets using a USP apparatus II (Distek dissolution system 2100A; Markham, ON, Canada). For vDAO activity, samples of 1 mL were withdrawn after each 1 h during the test.

The SGF and SIF used to simulate human gastric and intestinal transit (GIT) were prepared as described by the USP 43-NF38, 2020. The SGF was prepared by dissolving NaCl (2 g) and 3.2 g of pepsin in deionized water, and then the pH was adjusted to 1.2 ± 0.1 with HCl (37% *w*/*w*) in 950 mL. The SIF was prepared as follows: potassium phosphate monobasic (6.8 g), 77 mL 0.2 M NaOH, and 10 g pancreatin were dissolved in 750 mL of deionized water, and then the pH was adjusted to 6.8 ± 0.1 with 1 M NaOH. The final volume of each fluid was adjusted to 1000 (mL) with deionized water. The vDAO activity from the tablets in suspension in the simulated gastric and intestinal fluids was evaluated by DCHBS-AAP-HRP, as previously described in Section 3.3.

### 3.8. Statistical Analysis

All experiments used a minimum of three replicates. Data are expressed as the mean ± SD. Statistical tests were performed with GraphPad Prism software, using two-way ANOVA multiple comparisons. Differences were deemed statistically significant when the associated P-value was lower than 0.05.

## 4. Conclusions

The stability of DAO over time depends on storage conditions: temperature and humidity. The present study revealed the potential of disaccharides (sucrose and trehalose) as cryoprotectants to preserve vDAO activity during lyophilization and during long-term storage. The two cryoprotectants, SUC and TRE, have shown a similar effect on vDAO activity during freeze-drying and storage. The formation of the carbohydrate matrix, including the vDAO, seems to allow better preservation of the enzyme conformation by avoiding moisture absorption and preventing the decrease of enzyme activity due to variations in environmental conditions. When formulated as monolithic tablets, the vDAO preparations containing CPAs exhibited better resistance in simulated gastric and intestinal fluids and a better release pattern in intestinal fluid. These findings may be useful for further formulations of vDAO for the treatment of histamine-related dysfunctions.

## Figures and Tables

**Figure 1 molecules-28-00992-f001:**
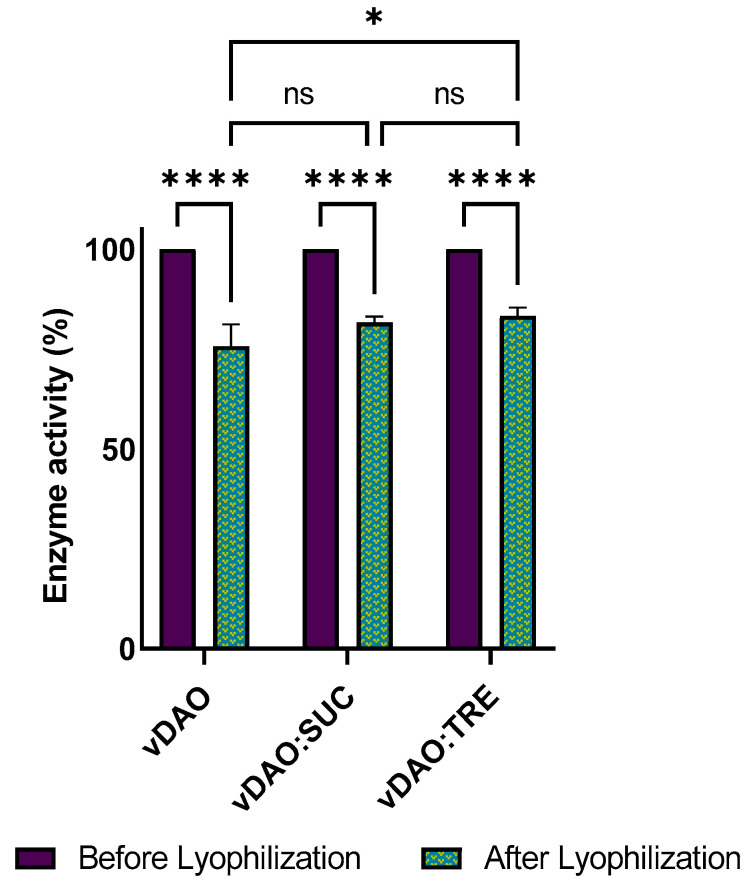
Effect of lyophilization on vDAO enzymatic activity in the presence or absence of sucrose (SUC) or trehalose (TRE) as cryoprotectants. vDAO activity was assayed by the DCHBS-AAP-HRP method. Reported values are the mean ± standard deviation (SD); *n* =3 different experiments; * *p* < 0.001, **** *p* < 0.0001, and ns: not significant; two-way ANOVA multiple comparisons.

**Figure 2 molecules-28-00992-f002:**
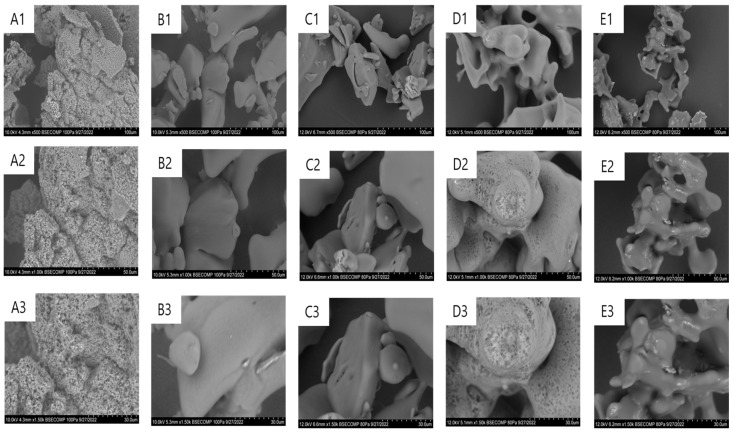
The morphology of vegetal diamine oxidase (vDAO) lyophilized samples: (**A**) vDAO, (**B**) with sucrose—vDAO-SUC, (**C**) with trehalose—vDAO-TRE, (**D**) Sucrose (SUC) alone, and (**E**) Trehalose (TRE) alone; SEM magnification (1) ×500, (2) ×1000, and (3) ×1500.

**Figure 3 molecules-28-00992-f003:**
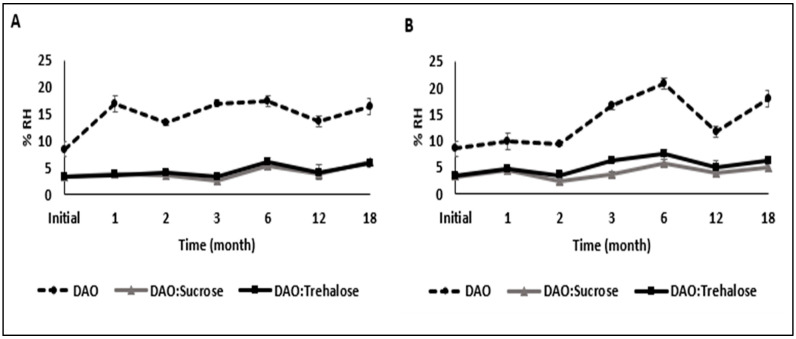
Time course of the residual humidity of vDAO powder formulations stored at different temperatures: (**A**) at 4 °C; (**B**) at −20 °C. The reported values are the mean ± standard deviation (SD); *n* = 3 different experiments.

**Figure 4 molecules-28-00992-f004:**
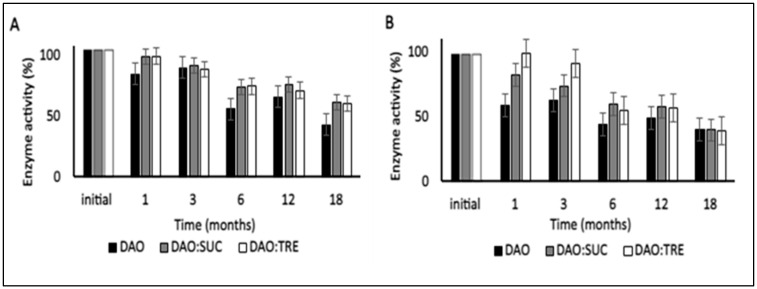
Effect of storage temperature on normalized vDAO activity stored at (**A**) 4 °C and (**B**) at −20 °C. vDAO activity was assayed by the DCHBS-AAP-HRP method. Reported values are the mean ± standard deviation (SD); *n* = 3 different experiments.

**Figure 5 molecules-28-00992-f005:**
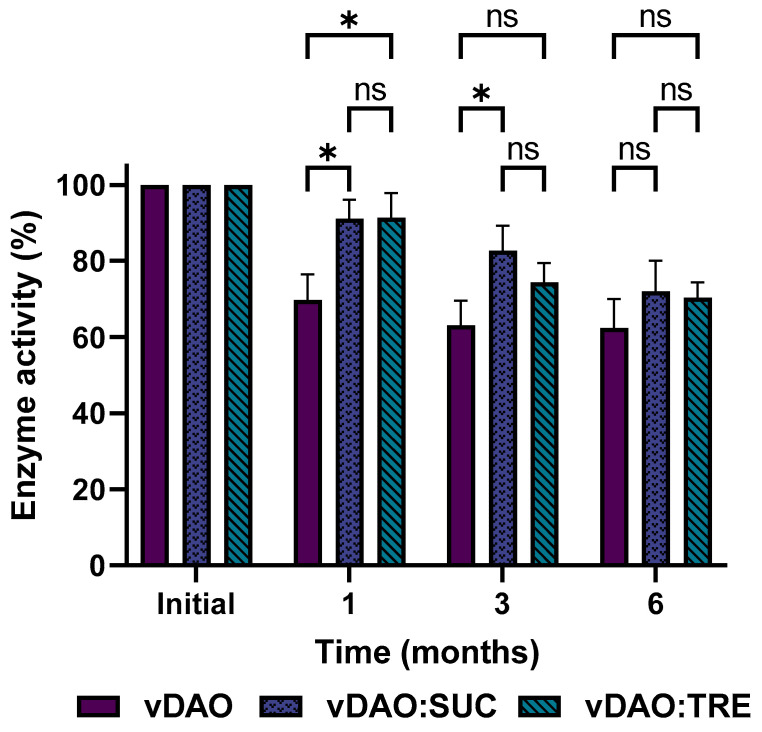
Enzymatic activity of vDAO in dry tablets of vDAO lyophilized powder with or without disaccharides as cryoprotectants during 6-months storage at 4 °C. vDAO activity was assayed by the DCHBS-AAP-HRP method. (Mean ± SD; *n* = 3; * *p* < 0.05 and ns: not significant; two-way ANOVA multiple comparisons.)

**Figure 6 molecules-28-00992-f006:**
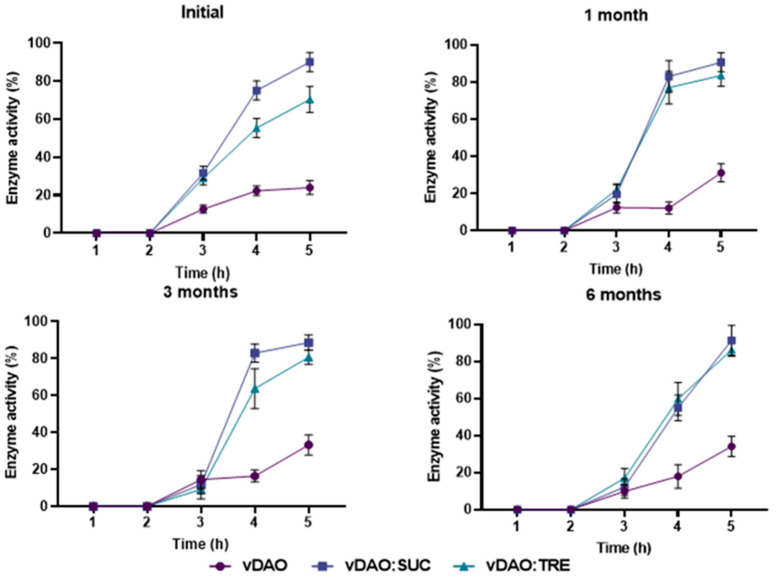
Release profiles of vDAO with and without cryoprotectants from monolithic tablets stored for 6 months at 4 °C. For the dissolution patterns, tablets were incubated at 37 °C 100 rpm in SGF (2 h) followed by SIF (3 h). At the indicated times, aliquots were withdrawn, and vDAO activity was determined by the DCHBS-AAP-HRP method. Reported values of vDAO activity refer to the enzymatic activity in the incubation media (mean ± SD; *n* = 3 different experiments).

**Figure 7 molecules-28-00992-f007:**
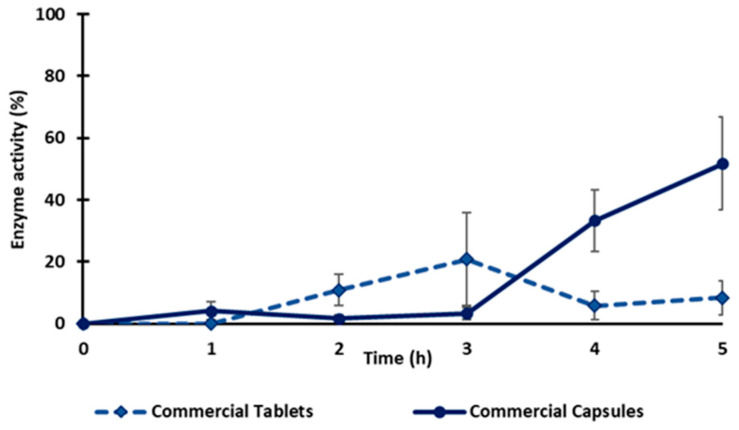
Release profile of vDAO from two commercial products (tablet and capsule forms). Tablets and capsules were incubated separately for 2 h in SGF (37 °C, 50 rpm) and then transferred to SIF. The activity of vDAO was measured every hour until complete dissolution by the DCHBS-AAP-HRP method. Reported values of vDAO activity refer to the enzymatic activity in the incubation media (mean ± SD; *n* = 3 different experiments).

**Table 1 molecules-28-00992-t001:** Thermal properties of vDAO lyophilized in the absence or the presence of cryoprotectants.

Formulation	T_g_ (°C) ^1^	T_c_ (°C) ^1^	Residual Humidity ^1^ (%)
vDAO	41.5 ± 7.7	152.1 ± 3.1	8.45 ± 0.82
vDAO:SUC	22.4 ± 3.1	n/a	3.27 ± 0.65
vDAO:TRE	34.9 ± 4.1	n/a	3.26 ± 0.71

^1^ T_g_ (glass transition), T_c_ (crystallization temperature), and RH (residual humidity, % *w*/*w*). All experiments were conducted in triplicate; the reported values are the mean ± standard deviation (SD).

**Table 2 molecules-28-00992-t002:** Summary of the physical properties of the tablets.

Formulation	Weight (mg)	Dimensions of Dry Tables		Hardness * (N)	Time for Complete Dissolution in SIF **
		Diameter * (mm)	Thickness * (mm)		
vDAO	300	9.60 ± 0.01	3.37 ± 0.03	12.03 ± 1.10	4–5 h
vDAO:Sucrose	300	9.59 ± 0.01	3.37 ± 0.03	12.13 ± 0.50	4–5 h
vDAD:Trehalose	300	9.61 ± 0.02	3.36 ± 0.03	9.53 ± 0.47	4–5 h

* All experiments were conducted in triplicate; the reported values are the mean ± standard deviation (SD). ** Time for Complete Dissolution in Simulated Intestinal Fluid (SIF), after 2 h in Simulated Gastric Fluid (SGF).

## Data Availability

Not applicable.

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
