# Peer review of "Enhanced Stability of Vegetal Diamine Oxidase with Trehalose and Sucrose as Cryoprotectants: Mechanistic Insights"

_molecules, 2023, doi:10.3390/molecules28030992_

Round 1
Reviewer 1 Report
The manuscript presents the enhanced stability of vegetal diamine oxidase with trehalose and sucrose as cryoprotectants and stresses on the mechanistic insights. It is meaningful, but improvement is necessary.
Statistical analysis in each experimental measurement should be added. Is there a significant difference among the enzyme activity values in Figures 1, 4, 5, 6, 7? How many is the parallel test. The *, **, or *** marks among the samples should be given.
The tablets obtained with powder of vDAO lyophilized with sucrose or trehalose cryoprotectants offered better protection for the enzyme activity. How is the difference between the potential of sucrose and of trehalose cryoprotectants to protect the enzyme from freeze-drying stress and during storage? It is suggested to give more discussion about this point.
Language should be polished by a native English speaker.
Author Response
MOLECULES - Section “Medicinal Chemistry”
Thanks are due for the comments and recommendations of the reviewer - this definitely helped to improve the manuscript. The paper was revised following the suggestions.
Response to Reviewer 1 Comments
The manuscript presents the enhanced stability of vegetal diamine oxidase with trehalose and sucrose as cryoprotectants and stresses on the mechanistic insights. It is meaningful, but improvement is necessary.
Point 1: Statistical analysis in each experimental measurement should be added. Is there a significant difference among the enzyme activity values in Figures 1, 4, 5, 6, 7? How many is the parallel test. The *, **, or *** marks among the samples should be given.
Response 1: A statistical analysis was performed on figures 1, 5 and the *, **, or *** marks among the samples were added as requested.
Note that in this study, all experiments were done in a minimum of three replicates (n=3). Data were expressed as the mean ± SD. Statistical tests were performed with the GraphPad Prism software, using Two-way ANOVA multiple comparisons.
More explicative sentences have been added to the Material and Method section, and also a note was added under each figure to mention the number of repetitions and the details of the *, ****, or ns marks, in order to facilitate the reading and understanding of the results.
Point 2: The tablets obtained with powder of vDAO lyophilized with sucrose or trehalose cryoprotectants offered better protection for the enzyme activity. How is the difference between the potential of sucrose and of trehalose cryoprotectants to protect the enzyme from freeze-drying stress and during storage? It is suggested to give more discussion about this point.
Response 2: First of all, as a general observation - there are no major differences in our study between trehalose and sucrose as cryoprotectants. Both of them afford a better stability after 18 months in powder forms. However, a difference was noticed in terms of the release time of enzymes from tablets. More precisely, tablets with vDAO:SUC generated a slightly higher percentage of liberation than those with vDAO:TRE. This behavior can be explained by the higher solubility of sucrose (200g/100mL) compared to that of trehalose (68.9g/100mL). Consequently, it may behave as a kind of "swelling" accelerator, favoring the enzyme liberation.
Action taken: The following paragraph was added to the revised manuscript, (section 2.5.2, row 280)
The slightly higher percentage of vDAO liberation from tablets with vDAO:SUC when compared to tablets with vDAO:TRE can be explained by the higher solubility of sucrose (200g/100mL) compared to that of trehalose (68.9g/100mL). Consequently, sucrose may behave as a kind of "swelling" accelerator, favoring the enzyme liberation.
Point 3: Language should be polished by a native English speaker.
Response 3: A thorough revision of the English style was done.
Reviewer 2 Report
Type of manuscript: Article
Title: Enhanced stability of vegetal diamine oxidase with trehalose and
sucrose as cryoprotectants: Mechanistic insights
Concerning about this manuscript, I would like to recommend authors to check and/or to clarify the following points:
1. vDAO is a glycoprotein with an N-linked carbohydrate content(3-14%) in vegetal DAOs. Is the carbohydrate part of vDAO essential for the enzyme activity? If yes, then most of arguments for the stability of the enzyme in this study are totally rational. However, if vDAO lacked carbohydrate part and that with carbohydrate part have the same enzyme activity, that means, the carbohydrate part is not essential for the enzyme activity, then some interpretations of results in this article should be revised. I therefore would like to recommend that authors try to search out literatures concerning about this issue and add this point somewhere in the introduction part of this manuscript.
2. Authors wrote “usually a protein-to-sugar weight ratio of 1:1 is suitable for optimal stabilization. In this study, enzyme: sugar ratio calculated as w/w mg protein: mg disaccharide was intentionally increased 1:10.” What does “intentionally” stand for? Did you check systematically protein-to-sugar ratio from 1:1, 1:2, …….1;10 to find out the optimal condition foe enhancing the stability of vDAO?
3. There are many synonyms of this enzyme, for instances diamine oxidase, histaminase, amine oxidase. I would like to recommend that authors unify the name of the enzyme as vDAO, if possible. I have pointed them out as follows;
Examples: On the 149th line, on the 199th line diamine oxidase can be replaced as vDAO. On the 201st line and on the 203rd line of DAO can be replaced as vDAO. On the 206th line, on the 215th line, and on the 254th line DAO can be substituted as vDAO. On the 188th line: The section 2.4 DAO…. can be replaced as vDAO. On the 354th line: “DAO activity expressed in enzymatic units (U); 1 unit corresponds to 1 μmol of H2O2 generated/min” can be changed as vDAO activity……… On the 415th line: “The stability of the DAO over time depends on the storage conditions:” can be changed as The stability of the vDAO over time depends on the storage conditions. On the 199th line: “Diamine oxidase is a glycoprotein with an N-linked carbohydrate content (3-14 % in vegetal AOs) [52].” Comment: I would like to recommend authors can check the correctness of vegetal AOs. Can it be vegetal DAOs?
4. On the 255th line: 2.5.2. In vitro dissolution tests can be changed as 2.5.2 In vitro dissolution tests. On the 301st line: “3.7.2. In vitro dissolution tests.” can be changed as 3.7.2. In vitro dissolution tests.
5. On the 105th line: “The enzyme catalytic activity was a control parameter to evaluate the preparations either in powder and tablet form was assayed during the storage period.”
Comments: either in powder and tablet can be changed as either in powder or tablet. Otherwise it can also be changed as both in powder and tablet.
6. On the 140th line “All experiments were done in triplicate. Glass transition (Tg), crystallization temperatures (Tc), and Residual humidity (% w/w).” can be changed as Crystallization temperature (Tc), and Residual humidity(RH; % w/w)
7. On the 197th and 198th line: “The moisture content may be related to eventual remaining traces of salts (i.e., Phosphates) used for the vDAO extraction procedure and that can retain humidity.” On the 325th line: “Vegetal Diamine oxidase (Diamaze) extracted from Pisum sativum (from IBEX Pharmaceuticals Inc., Montreal, QC, Canada) was kept at -80 °C.”
Comments: I would like to recommend authors to describe briefly the vDAO extraction procedure and also to insert either the specific activity of vDAO or total activity of vDAO after this extraction of vDAO from Pisum sativum (from IBEX Pharmaceuticals Inc., Montreal, QC, Canada). Authors can also represent the SDS polyacrylamide gel of this extract to show the purity of the vDAO after the extraction state.
8. On the 336th line: “The ratio of amine oxidase:cryoprotectant was 1:10 (w:w).” can be changed as The ratio of vDAO : cryoprotectant was 1:10 (w/w).
9. On the 399th line: “The solution was filtered; the vDAO activity was determined using the method described in section 3.”
Comments: I would like to recommend authors should give the pore size of filter. Did authors not have any problems by filtration because of carbohydrate part of vDAO? Please describe it briefly if they had some problems by filtration.
Author Response
MOLECULES - Section “Medicinal Chemistry”
Thanks are due for the comments and recommendations of the reviewer - this definitely helped to improve the manuscript. The paper was revised following the suggestions.
Response to Reviewer 2 Comments
Point 1: vDAO is a glycoprotein with an N-linked carbohydrate content(3-14%) in vegetal DAOs. Is the carbohydrate part of vDAO essential for the enzyme activity? If yes, then most of arguments for the stability of the enzyme in this study are totally rational. However, if vDAO lacked carbohydrate part and that with carbohydrate part have the same enzyme activity, that means, the carbohydrate part is not essential for the enzyme activity, then some interpretations of results in this article should be revised. I therefore would like to recommend that authors try to search out literatures concerning about this issue and add this point somewhere in the introduction part of this manuscript.
Response 1: Yes, the carbohydrate part is essential for the enzyme activity and for the stabilization of the enzyme conformation. In fact, it has been shown with vegetal DAO from lentils legume that the carbohydrate moiety is essential for the enzyme activity (Moosavi et al, 2007) and for its stabilization. Additional nonreducing sugars would probably generate some more stabilization by carbohydrate:carbohydrate interactions via hydrogen bonding. This aspect was discussed in the 2.3 section.
The reference mentioned above is found as a citation in the body of the manuscript and to the list of references.
Point 2: Authors wrote “usually a protein-to-sugar weight ratio of 1:1 is suitable for optimal stabilization. In this study, enzyme: sugar ratio calculated as w/w mg protein: mg disaccharide was intentionally increased 1:10.” What does “intentionally” stand for? Did you check systematically protein-to-sugar ratio from 1:1, 1:2, …….1;10 to find out the optimal condition for enhancing the stability of vDAO?
Response 2: Previous studies of protein stabilization by sucrose and trehalose reported that the best ratio was 1:10 (Ref. : Carpenter, et al 2002). In addition, several experiments have been carried out in our laboratory on vDAO with different types of carbohydrates (mono, and disaccharides including sucrose and trehalose) with different ratios 1:1, 1:5 (Rezaei, 2020). The study was limited to a three-month period and the retained enzyme activity was 62-66 %, whereas in the current study, with the 1:10 ratio, the retained activity after three months was higher: about 90 %. This is why we continued the stability study with a 1:10 protein:cryoprotectant ratio.
Action taken: The reference [39] (Carpenter, et al 2002) was assigned to the concerned paragraph (Results and Discussion section).
Point 3: There are many synonyms of this enzyme, for instance diamine oxidase, histaminase, amine oxidase. I would like to recommend that authors unify the name of the enzyme as vDAO, if possible. I have pointed them out as follows;
Examples: On the 149th line, on the 199th line diamine oxidase can be replaced as vDAO. On the 201st line and on the 203rd line of DAO can be replaced as vDAO. On the 206th line, on the 215th line, and on the 254th line DAO can be substituted as vDAO. On the 188th line: The section 2.4 DAO…. can be replaced as vDAO. On the 354th line: “DAO activity expressed in enzymatic units (U); 1 unit corresponds to 1 μmol of H2O2 generated/min” can be changed as vDAO activity……… On the 415th line: “The stability of the DAO over time depends on the storage conditions:” can be changed as The stability of the vDAO over time depends on the storage conditions. On the 199th line: “Diamine oxidase is a glycoprotein with an N-linked carbohydrate content (3-14 % in vegetal AOs) [52].” Comment: I would like to recommend authors can check the correctness of vegetal AOs. Can it be vegetal DAOs?
Response 3: All synonyms were changed according to the recommendations. Vegetal AOs correctness was checked and it could be applied to vDAO. The references [Moosavi et al., 2007; Meda et al. 1997] have been added to support this assertion.
Action taken: All synonyms have been now reduced to vDAO as a unique abbreviation. An additional reference [53] was added to support the information on the carbohydrate moiety of vDAO.
Point 4: On the 255th line: 2.5.2. In vitro dissolution tests can be changed as 2.5.2 In vitro dissolution tests. On the 301st line: “3.7.2. In vitro dissolution tests.” can be changed as 3.7.2. In vitro dissolution tests.
Response 4: The font style has been changed according to the recommendations.
It was an error in the conversion to PDF.
Action taken – we will try to solve this point
Point 5: On the 105th line: “The enzyme catalytic activity was a control parameter to evaluate the preparations either in powder and tablet form was assayed during the storage period.”
Comments: either in powder and tablet can be changed as either in powder or tablet. Otherwise, it can also be changed as both in powder and tablet.
Response 5: “The enzyme catalytic activity was a control parameter to evaluate the preparations in both powder and tablet form was assayed during the storage period.”
Action taken: The sentence was amended as recommended.
Point 6: On the 140th line “All experiments were done in triplicate. Glass transition (Tg), crystallization temperatures (Tc), and Residual humidity (% w/w).” can be changed as Crystallization temperature (Tc), and Residual humidity(RH; % w/w).
Response 6: The text was amended as suggested.
Point 7: On the 197th and 198th line: “The moisture content may be related to eventual remaining traces of salts (i.e., Phosphates) used for the vDAO extraction procedure and that can retain humidity.” On the 325th line: “Vegetal Diamine oxidase (Diamaze) extracted from Pisum sativum (from IBEX Pharmaceuticals Inc., Montreal, QC, Canada) was kept at -80 °C.”
Comments: I would like to recommend authors to describe briefly the vDAO extraction procedure and also to insert either the specific activity of vDAO or total activity of vDAO after this extraction of vDAO from Pisum sativum (from IBEX Pharmaceuticals Inc., Montreal, QC, Canada). Authors can also represent the SDS polyacrylamide gel of this extract to show the purity of the vDAO after the extraction state.
Response 7: The vDAO used in this study was a commercial product, and the method of extraction wasn’t disclosed by the supplier. A certificate of analysis was provided by the manufacturer with the quality control specification notifying the enzyme activity. Moreover, an “in-house” electrophoretic zymographic pattern of enzyme activity and the SDS electrophoretic pattern of proteins completed the characterization of the received product. The electrophoresis profile showed a satisfactory degree of purification.
Action taken: The zymographic and SDS-PAGE electrophoretic patterns have been added as supplementary material (Figure S2).
Point 8: On the 336th line: “The ratio of amine oxidase:cryoprotectant was 1:10 (w:w).” can be changed as The ratio of vDAO : cryoprotectant was 1:10 (w/w).
Response 8: Changed as recommended.
Point 9: On the 399th line: “The solution was filtered; the vDAO activity was determined using the method described in section 3.”
Comments: I would like to recommend authors should give the pore size of filter. Did authors not have any problems by filtration because of carbohydrate part of vDAO? Please describe it briefly if they had some problems by filtration.
Response 9: The solution was filtered on filter paper (FisherBrand Filter paper, medium porosity) with no problems during the filtration.
Action taken: This information was added to the Materials and Methods section as follows: The vDAO solutions were filtered on filter paper (FisherBrand Filter paper, medium porosity).
Round 2
Reviewer 1 Report
Authors have revised the manuscript carefully by considering the original comments. It could be accepted for publication in the journal of Molecules.